# A Peer-review Look on Multi-modal Clustering: An Information Bottleneck Realization Method

Zhengzheng Lou [1]   Hang Xue [1]   Chaoyang Zhang [1]   Shizhe Hu [1]

## Abstract

Despite the superior capability in complementary information exploration and consistent clustering structure learning, most current weight-based multi-modal clustering methods still contain three limitations: 1) lack of trustworthiness in learned weights; 2) isolated view weight learning; 3) extra weight parameters. Motivated by the peer-review mechanism in the academia, we in this paper give a new peer-review look on the multi-modal clustering problem and propose to iteratively treat one modality as "author" and the remaining modalities as "reviewers" so as to reach a peer-review score for each modality. It essentially explores the underlying relationships among modalities. To improve the trustworthiness, we further design a new trustworthy score with a self-supervision working mechanism. Following that, we propose a novel Peer-review Trustworthy Information Bottleneck (PTIB) method for weighted multi-modal clustering, where both the above scores are simultaneously taken into account for accurate and parameter-free modality weight learning. Extensive experiments on eight multi-modal datasets suggest that PTIB can outperform the state-of-the-art multi-modal clustering methods.

## 1. Introduction

Learning a consistent clustering result from different modalities by fully mining the underlying modality correlations is the essence of multi-modal clustering (MMC) (Raya et al., 2024). From the view of basic method, existing MMC methods are generally based on $k$-means, canonical correlation analysis, spectral clustering, matrix factorization, information bottleneck and deep learning methods. From the perspective of strategies, current MMCs can be classified into weighted MMC (Wang et al., 2020a), shared feature subspace learning based MMC (Li et al., 2022), multi-modal consensus clustering (Liu et al., 2021; Liang et al., 2024), multi-modal co-clustering (Kumar et al., 2011; Zhang et al., 2024), multi-modal subspace clustering (Li et al., 2022), and tensor representation learning based MMC (He & Atia, 2022; Gu et al., 2024). With the rapid development of various kinds of methods, MMC has been successfully applied into many practical applications, such as object recognition (Lou et al., 2013), human action recognition (Hu et al., 2020) and coherent groups detection in crowd scenes (Wang et al., 2020b).

**Related Works.** Investigated from the above methods, the weighted MMCs have exhibited remarkable clustering performance in the past decade, which is mainly due to its superior capability in complementary relationship discovery and consistent clustering structure learning. For the example of some typical ones, Xu et al. (2016) incorporated feature selection into weighted MMC for solving the high-dimensional data clustering, so that the discriminative representations and accurate clustering results can be jointly learned. Further, to address the challenging large-scale data clustering, Zhang et al. (2019) proposed an efficient binary MMC method, worked by collaboratively conducting the discrete representation learning and binary clustering assignment discovery. Different from most weighted MMCs, Zhao et al. (2020) designed a cluster weights learning based method to find the more fine-grained cluster relationships compared to the modality-weighted methods. For the difficult parameter learning issue, Wang et al. (2020a) focused on automatically tuning the weight parameters to obtain a refined unified graph fusion matrix. In more recent year, Zhou & Shen (2020) leveraged adversarial learning and attention mechanism to align the latent feature distributions and quantify the importance of modalities respectively. Xia et al. (2023b) introduced an adaptive fusion layer to adaptively sense the importance of modalities. Hu et al. (2025) introduced a simple fusion mechanism that dynamically updates modality-specifc weights via backpropagation, similar to parameter optimization. Chen et al. (2023) explored the higher-order relations across modalities by designing a low-rank Tensor based proximity learning method for MMC

---

[1]School of Computer Science and Artificial Intelligence, Zhengzhou University, Zhengzhou, China. Correspondence to: Shizhe Hu <ieshizhehu@gmail.com, https://shizhehu.github.io/>.

*Proceedings of the 42$^{nd}$ International Conference on Machine Learning*, Vancouver, Canada. PMLR 267, 2025. Copyright 2025 by the author(s).

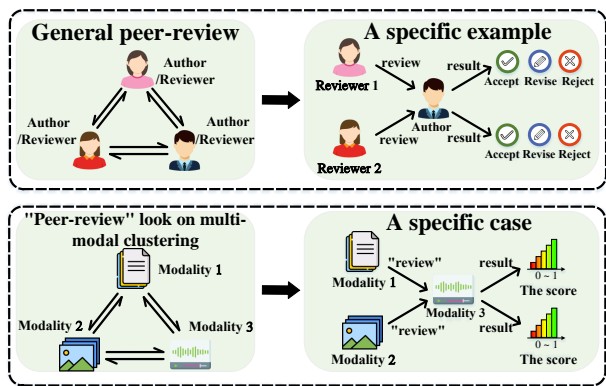

Figure 1. The upper box indicates the general peer-review process in academia, while the lower box shows the peer-review look on multi-modal clustering, where each modality can be either an "author" or a "reviewer".

problem.

**Limitations.** Investigated from the above closely-related weighted MMC works, we observed there mainly lies three limitations. First, all the existing weighted MMCs only focus on learning the modality weights using different methods, ignoring whether the learned weights are trustworthy or not. Second, the weight learning mechanisms of most weighted MMCs is separated and essentially relied on the "single-modal information", e.g., $w^i = \frac{1}{F^i}$, where $w^i$ and $F^i$ indicate the modality weight and the objective function value of the $i$-th modality respectively. Thus, the modality correlations are not fully explored and exploited, which may degrade the clustering performance. Third, most weighted MMCs require one or more parameters for controlling the weight distribution, where these parameters are usually difficult to be tuned by hand without any prior knowledge.

**Motivation.** In academia, peer-review mechanism is always adopted to mutually assess the contribution of one specific work to the community, discuss the significance of work, and provide communication to peers. Similarly, different modalities jointly contribute to the multi-modal clustering task, making it necessary to assess their individual contributions. Inspired by this, we naturally give a peer-review look on the multi-modal clustering problem. As illustrated in Figure 1, in general peer-review process, a person may be an author or a reviewer. When a person acts as an author, the work will be assigned to different reviewers by Editor in Chief (EIC) / Associate Editor (AE) to provide recommendations and suggestions. The peer-review mechanism facilitates the improvement and refinement of work, ensuring high-quality and reliable publications. From the "peer-review" look on multi-modal clustering, one modality can either be an "author" or a "reviewer". The "reviewer" modalities review the work of the "author" modality, and produce feedback review scores to evaluate the contribution

or quality of the "author" modality. By mimicking this interesting mechanism in Figure 1, we can employ the feedback review scores to effectively integrate the complementary discriminative information across modalities so as to promote multi-modal clustering performance.

**Contribution.** In this paper, we propose to address the multi-modal clustering problem from the new peer-review look. It allows the remaining modalities as "reviewers" to review one "author" modality, and provides peer-review scores for each modality. Additionally, the corresponding trustworthy score is designed to measure the reliability of different "reviewer" modalities, thus ensuring the trustworthiness of the peer-review scores learned from the "peer-review" process. With the remarkable performance of the popular information bottleneck theory (Tishby et al., 1999) in multi-modal learning (Gao et al., 2007; Lou et al., 2013; Hu et al., 2022), we in this paper propose a Peer-review Trustworthy Information Bottleneck (PTIB) method for solving the weighted multi-modal clustering problem. The clustering result obtained in each iteration is used to update the trustworthy score of each "reviewer" modality, working in a self-supervised mechanism. Thus, a more reasonable trustworthy evaluation of the peer-review score for each modality is reached. The peer-review scores given by "reviewer" modalities and the corresponding trustworthy scores for evaluating these "reviewer" modalities are combined to determine the importance or contribution (using modality weights in this paper) of each "author" modality. Moreover, we solve the optimization problem of PTIB by an effective $k$-means-like algorithm. Experiments on eight multi-modal datasets demonstrate the superiority and effectiveness of the proposed method.

The major contributions are summarized as follows:

- A novel peer-review trustworthy information bottleneck (PTIB) method is proposed for multi-modal clustering, which performs by jointly learning the peer-review scores on different modalities and the trustworthy scores for quantifying the trustworthiness of the peer-review scores.

- We give a new peer-review look on the multi-modal clustering problem, and thus design a peer-review score for evaluating the quality of each modality.

- A corresponding trustworthy score is newly designed to evaluate the trustworthiness of peer-review score, ensuring the reliability of multi-modal peer-review.

- Rich experiments on eight multi-modal datasets demonstrate the superiority and effectiveness of the proposed PTIB.

**Relations Between Multi-modal and Multi-view Clustering.** Generally, the aim of multi-modal clustering and

multi-view clustering is similar, especially in integrating different sources of information for improving clustering performance. The differences between them are as follows: Multi-view learning focuses on diverse feature representations of the same object, while multi-modal learning deals with complex relationships between heterogeneous modalities, which is more complicated to handle. In practical applications, overlaps between them may exist (e.g., multi-modal data can also be considered as generalized multi-view data). However, technical solutions should be selected based on data characteristics (feature homogeneity and semantic consistency) to ensure methodological compatibility. In this paper, we use a more general expression of multi-modal clustering instead of multi-view clustering.

## 2. Revisit: Information Bottleneck

Information bottleneck (IB) principle (Tishby et al., 1999) originated from the rate-distortion theory, and its details can be referred from our survey (Hu et al., 2024b).

For clustering problem, IB considers it as a data compression process, which attempts to learn an optimal compressed representation $T$ of $X$ while maximally maintaining the relevant information about variable $Y$. It is formally described as

$$R(D) = \min_{\{p(t|x):I(T,Y)\geq D\}} I(T;X), \qquad (1)$$

where the compact representation $T$ compresses the source variable $X$ while maximally capturing the relevant information with respect to the variable $Y$, $p(t|x)$ indicates the probability of data point $x$ being assigned to $t$-th cluster, $I(T;X)$ and $I(T;Y)$ denote the mutual information between the compressed variable $T$ and variable $X$ and $Y$ respectively.

By adopting a positive $\beta$, we have the following Lagrange version of the IB method

$$\mathcal{L}_{min}[p(t|x)] = I(T;X) - \beta I(T;Y), \qquad (2)$$

where $\beta \in (0, +\infty)$ is a Lagrange multiplier which serves as a trade-off parameter between data compression and relevant information preservation. And a formal iterative solution for the above Eq. (2) is given as

$$p(t|x) = \frac{p(t)}{Z(x,\beta)} e^{-\beta D_{KL}[p(y|x)||p(y|t)]}, \qquad (3)$$

where $p(t) = \sum_x p(t|x)p(x)$, $Z(x,\beta)$ is a normalization function, $p(y|t) = \frac{1}{p(t)} \sum_x p(t|x)p(x,y)$, and $D_{KL}$ (Cover & Thomas, 2006) is the *Kullback-Leibler* divergence.

In recent years, IB theory has been widely used in various multi-modal clustering tasks. For example, Federici et al. (2020) propose a multi-modal IB method that can identify non-shared information between two modalities. Yan et al.

(2024) propose a multi-modal IB method that uses shared representations of multiple modalities to eliminate private information of a single modality. But the modality-private information is eliminated as much as possible during the process of data compression, only exploring the shared information of modalities without taking advantage of the complex relationship between modalities. Hu et al. (2024a) learn embeddings on two distinct feature spaces, reconstruct semantic information in a parallel manner, and IB theory is further used to reduce representation noise. However, its final clustering result is obtained by directly averaging the local clusters from the modal high-dimensional features.

Different from the existing multi-modal clustering methods based on IB theory, the proposed method considers the complex relationship between modalities, where the designed multi-modal peer-review process is used to reasonably score the contribution of each modality, and the trustworthiness of it is ensured in a self-supervised manner.

## 3. The Proposed Method

In this section, we first give the problem formulation and overall framework, then elaborately picture the details of the PTIB method.

**Problem Formulation.** Given the random variable $X = \{x_1, x_2, \ldots, x_n\}$ denoting the set of $n$ data samples from the dataset $\mathcal{X}$, we have the random variable $\{Y^j\}_{j=1}^m$ denoting the data feature representation of $m$ different modalities, where $Y^j \in R^{n*d^j}$ indicates that the feature dimension of the $j$-th modality is $d^j$. Then, we reach the corresponding co-occurrence matrix $\{X, Y^j\}_{j=1}^m$ and its joint probability distribution $\{p(X, Y^j)_{j=1}^m\}$ by adopting the popular Bag-of-Words model (Fei-Fei & Perona, 2005).

**Overall Framework.** As the overall framework illustrated in Figure 2, PTIB first captures the local clustering structure of each modality, i.e., $\{T^1, T^2, \cdots, T^m\}$, with the IB method, and then obtains the peer-review scores for each "author" modality with the evaluation of remaining "reviewer" modalities. Meanwhile, the final clustering result is adopted for assessing the trustworthiness of each modality in a self-supervision fashion. With the peer-review $\{\mu^1, \mu^2, \cdots, \mu^m\}$ and trustworthy $\{\sigma^1, \sigma^2, \cdots, \sigma^m\}$ score for each modality, more accurate modality weights and improved clustering result are learned in each iteration.

Regarding the trustworthiness of multiple modalities, almost all existing methods focus on trustworthy multi-modal classification(Han et al., 2021; 2023; Zheng et al., 2023; Zou et al., 2023). Han et al. (2023) introduce the variational Dirichlet to characterize the distribution of the class probabilities, parameterized with evidence from different views and integrated with the Dempster-Shafer theory, thus promoting both classification reliability and robustness. Zheng

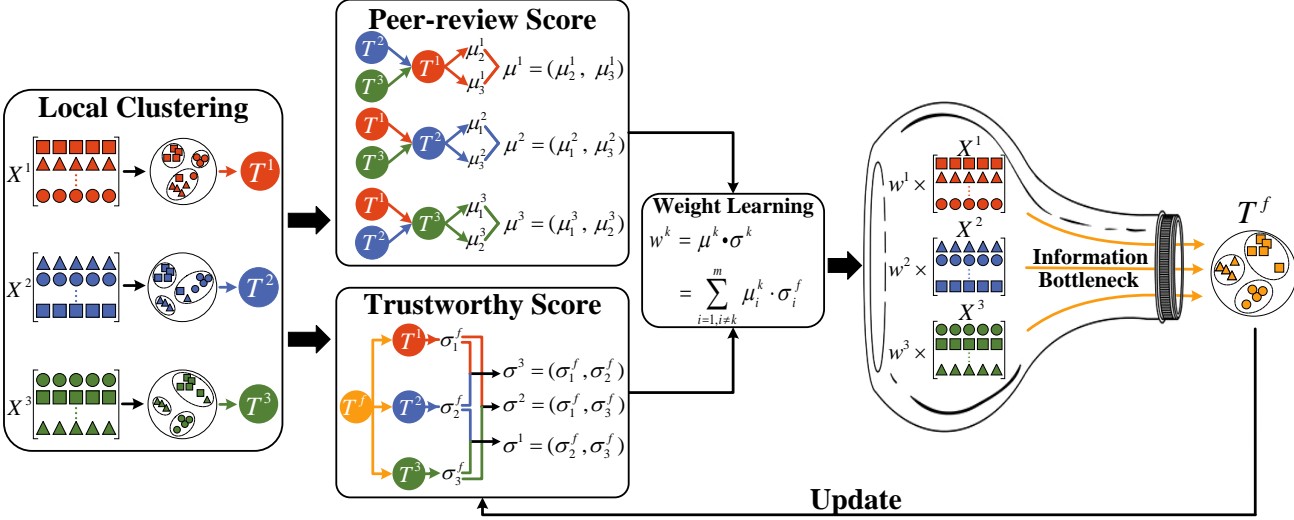

*Figure 2.* The framework of PTIB. Each modality is treated as an "author" or "reviewer", with its local clustering result serving as the "work" or "criteria". A peer-review is conducted in the multi-modal setting to obtain peer-review score. Additionally, the final clustering result played "EIC /AE" quantifies the reliability of "reviewer" modality to obtain trustworthy score. Both the two scores are taken into account for modality weight learning, leading to more accurate weights and improved clustering result through iteration.

et al. (2023) proposes a trustworthy multi-modal classification network via multi-level confidence learning, which integrates both feature and label-level confidence learning for trustworthy multi-modal classification. Zou et al. (2023) induces a transparent fusion strategy based on the modality confidence estimation strategy to track information variation within different modalities for dynamical fusion. Different from them, the proposed method aims to guarantee the trustworthiness of the learned modal weights in a self-supervised manner. To the best of our knowledge, none of the existing weighted MMCs employ the trustworthy strategy in the weight learning process.

### 3.1. Peer-review Score

Here, we give a new peer-review look on the multi-modal clustering, resorting the advantages of this mechanism to learn modal weight through modal interaction.

In a general peer-review process, there are often some specific review criteria conducted to assess and score the author's work. By mimicking this, we reach the peer-review score in a similar manner under the multi-modal peer-review process. First of all, it is necessary to establish the peer-review criteria for different reviewer modalities. The feature of each modality contains unique characteristics, but their different feature dimensions make it hard to quantify their discrepancy. Therefore, we instead adopt the local clustering result of the reviewer modality as review criteria. Then, the author's work is also represented with the local clustering result, which can be directly compared with the review

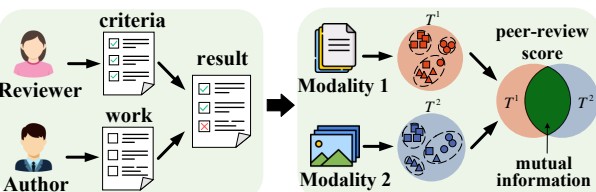

*Figure 3.* The local clustering results from the modality 1 and modality 2 are regarded as the criteria and the work, respectively. The peer-review score is the mutual information between them.

criteria. The peer-review score depends on how similar it is to the review criteria. Apparently, the general metric of mutual information is a good choice for quantifying them. For clarity, we give a two-modal example in Figure 3. To be more accurate, we adopt the normalized mutual information version and formally define it as follows

$$\mu_i^k = \frac{2 \times I(T^i, T^k)}{H(T^i) + H(T^k)} \qquad (4)$$

where $\mu_i^k$ is the peer-review score given by the reviewer modality $V^i$ to the author modality $V^k$, $T^i$ and $T^k$ are the local clustering results obtained from the reviewer modality $V^i$ and the author modality $V^k$, respectively. $H(\bullet)$ denotes the entropy of one variable. Naturally, all of the peer-review scores of author modality $V^k$ can be represented as a vector

$$\mu^k = \{\mu_1^k, \ldots, \mu_i^k, \ldots, \mu_m^k\}, i \neq k. \qquad (5)$$

**Remark.** Actually, there are many ways to attain the final peer-review scores, while we only select the normalized mutual information as a typical metric. Additionally, although the peer-review score provides a quantitative analysis for exploring the modality correlations, it still has its limitations. Due to the varying quality of multiple modalities, low-quality "reviewer" modality may give an inaccurate peer-review score to a relatively high-quality "author" modality. This may probably lead to wrong evaluation of the importance of modalities and eventually do harm to the clustering performance. Hence, it is imperative to analyze and evaluate the trustworthiness of the peer-review scores given by reviewer modalities to ensure their reliability.

### 3.2. Trustworthy Score

In this part, we propose to regard the final clustering assignment as the "EIC / AE", which is then used to evaluate the trustworthiness of the peer-review score in each iteration with a self-supervision fashion. The final clustering assignment improves as the iteration increases, thus leading to more accurate trustworthy scores.

Clustering is to divide samples into different clusters according to their features, which is similar to a two-layer decision tree process. By treating the input sample set as the root node of decision tree, each leaf node thus corresponds to one specific cluster of the reviewer modality. Thus, we assess the trustworthiness of the reviewer modality by measuring cluster uncertainty, as defined follow, much like evaluating the quality of decision by measuring leaf purity.

**Definition 3.1** (**Major/Minor Category**). Given a multi-modal dataset, if the local clustering result of modality supervised by the final clustering result, the category of correctly assigned samples in a cluster of a specific local clustering result is called major category, the set of categories of incorrectly assigned samples in it is called minor categories.

**Definition 3.2** (**Cluster Uncertainty**). For one reviewer modality, if the probability of major category in a cluster is $p$, then the cluster uncertainty can be measured by the following information entropy

$$H(p) = -p \log_2 p - (1-p) \log_2(1-p). \quad (6)$$

The final clustering result cannot make a completely accurate judgment on the local clustering result like the true label. We can only expect that the two are as similar as possible, that is, the probability of major category is as larger as possible. In a self-supervised scenario, the mixing of minor categories introduces interference when judging the information of the current cluster, so we only use the $p$ and $1-p$ to represent the cluster uncertainty. The following theorem can prove this claim and its proof is given in Appendix A.1.

**Theorem 3.3.** *For arbitrary cluster of a reviewer modality, the more mixed the minor categories gain, the higher the*

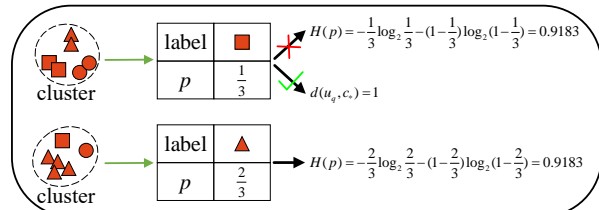

*Figure 4.* A typical example of the "symmetry crisis" issue that the poor and better clusters may have the same cluster uncertainty.

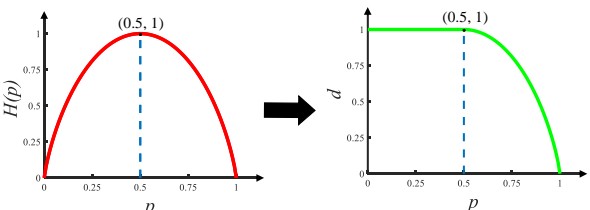

*Figure 5.* The solution of handling the "symmetry crisis" problem.

*entropy of it gets, thus resulting in high uncertainty of the cluster and interference in information judgment.*

Eq. (6) leads to an issue that a poor cluster and a better cluster obtain the same value of Eq. (6), where we call it "symmetry crisis". A typical example is illustrated in Figure 4. To address this crisis, we assume that a cluster with more than half incorrectly assigned samples (i.e., less than half major category) is of low quality, and we directly set its cluster uncertainty as the max value, as shown our handling of it in Figure 5. Finally, based on Definition 3.2, we define the following concept of cluster distortion to evaluate the quality of the reviewer modality.

**Definition 3.4** (**Cluster Distortion**). Let $U = \{u_1, u_2, \ldots, u_{|U|}\}$ denote the set of clusters in each reviewer modality and $C = \{c_1, c_2, \ldots, c_{|U|}\}$ denote the set of clusters in the final clustering result in each iteration, then the cluster distortion from $U$ to $C$ is given by

$$d(u_q, c_*) = \begin{cases} 1, & \text{if } 0 \leq p < \frac{1}{2}, \\ H(p), & \text{if } \frac{1}{2} \leq p \leq 1. \end{cases} \quad (7)$$

where $u_q$ $(1 \leq q \leq |U|)$ denotes the arbitrary cluster in the cluster set $U$, $c_*$ denotes the major category of each cluster $u_q$ in $U$, and $p = \frac{1}{|u_q|}|u_q \cap c_*|$. Note that $d(u_q, c_*) = 1$ indicates that a cluster dominated by incorrectly assigned samples have the largest distortion.

Based on the distortion measurement Eq. (7) with respect to a cluster, the distortion between $U$ and $C$ is defined as follows

$$D(U, C) = \frac{1}{|U|} \sum_{q=1}^{|U|} d(u_q, c_*). \quad (8)$$

The above clustering distortion works by measuring how much the local clustering result of a single reviewer modality distorts from the final clustering result, similar to the evaluation from EIC / AE to reviewer in peer-review mechanism. It is noted that this approach is coincidentally consistent with the self-supervised learning. Generally, the smaller the clustering distortion is, the more reliable the reviewer modality is. Formally, the trustworthy score is defined by

$$\sigma_i^f = \frac{1}{D(T^i, T^f)}, \quad (9)$$

where $\sigma_i^f$ is the trustworthy score of reviewer modality $V^i$, $T^i$ is the local clustering result of reviewer modality $V^i$, and $T^f$ is the final clustering result. Similarly, all the trustworthy scores for reviewer modalities of reviewing the author modality $V^k$ can be represented as a vector

$$\sigma^k = \{\sigma_1^f, \ldots, \sigma_i^f, \ldots, \sigma_m^f\}, i \neq k. \quad (10)$$

### 3.3. Modality Weight Learning

By jointly considering the peer-review and trustworthy score, we obtain the final modality weight with the inner product as follows

$$w^k = \mu^k \bullet \sigma^k = \sum_{i=1,i\neq k}^{m} \mu_i^k \cdot \sigma_i^f, m > 2. \quad (11)$$

Note that Eq. (11) is only applicable when there are more than two modalities. Next, we will discuss in detail the circumstance where two modalities are given in the following.

**Two-modal Prejudice Processing.** In academic peer-review, unreliable reviewers can give an unreasonable dislike or preference for the author's work. Similarly, in multi-modal setting, reviewer modalities tend to have modality prejudice. If we have more than two modalities, the modality prejudice will be eliminated by multiple reviewers and their trustworthiness in modality weight learning. However, it can not be solved like this if only two modalities.

Assume only two modalities, it is inevitable that one modality will be better or worse than another, so they may have different trustworthy scores. As shown in Figure 6(a), due to the single reviewer, the modality prejudice can not be eliminated, leading to unreasonable modality weight. Similar to the scenario that the EIC / AE makes the final decision by directly adopting his/her own review instead of using the comments from the unreliable reviewer, it can learn the modality weight by directly leveraging the trustworthy score given by the final clustering result to the author modality, as shown in Figure 6(b). In this case, the peer-review score has no effect, as they are all the same value due to the symmetric scoring function.

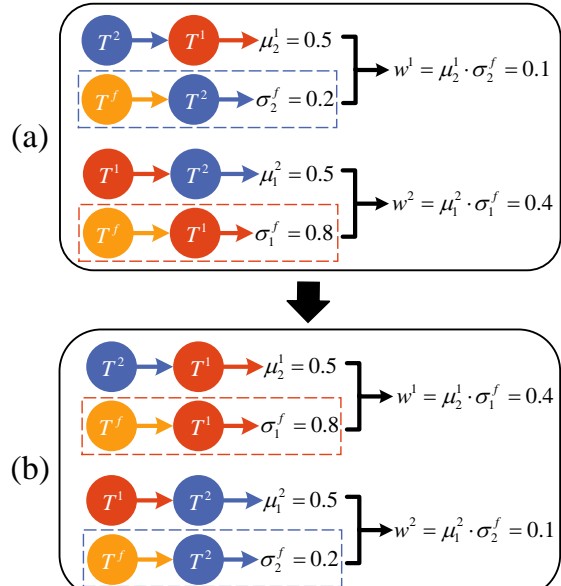

*Figure 6.* Example of two-modal prejudice processing. (a) It is unreasonable that high-quality modality 1 with high trustworthy score learned a small weight. (b) $T^f$ gives the final modality weights by directly adopting its own review.

Based on above, we summarize the final modality weight learning as follows

$$w^k = \begin{cases} \sum\limits_{i=1,i\neq k}^{m} \mu_i^k \cdot \sigma_k^f, & \text{if } m = 2, \\ \sum\limits_{i=1,i\neq k}^{m} \mu_i^k \cdot \sigma_i^f, & \text{if } m > 2. \end{cases} \quad (12)$$

### 3.4. The Objective Function

Finally, with the above modality weight learning mechanism, we have the objective function with the popular information bottleneck realization as follows

$$\mathcal{F}_{max}[p(t|x)] = \sum_{i=1}^{m} w^i \cdot [I(T; Y^i) - \beta^{-1} I(T; X)], \quad (13)$$

when $\beta \in (0, +\infty)$ balances the information compression and preservation. Note that $\beta$ equal to $+\infty$ represents the extreme compression, and in this case it is parameter-free.

The advantages and possible weaknesses of the proposed method are discussed in Appendix B.

### 3.5. Optimization Method

We solve the optimization problem of PTIB with an effective sequential $k$-means-like draw-and-merger algorithm (Lou et al., 2013; Hu et al., 2020; 2022), where each sample is sequentially drawn from the old cluster and assigned to

an optimal new cluster that minimizes the merger cost to maximize the objective function. It is shown as follows:

**Weight Initialization.** Modality weights are initialized with the initial peer-review and trustworthy scores.

**Random Clustering.** The initial input, i.e., $X$, is randomly partitioned to $|T|$ data clusters.

**Draw.** Each sample $x$ is sequentially drawn from its "old" cluster $t^{old}$ of the $i$-th modality, which is then taken as a separate cluster $\{x\}$. Now, it leads to $|T| + 1$ data clusters.

**Merger.** To make the number of clusters recover to $|T|$, a "new" cluster $t^{new}$ is required to be selected from the existing data clusters for merger, which meanwhile satisfies the minimal cost for the separate cluster $\{x\}$. This ensures that the separate cluster $\{x\}$ will be merged to the optimal candidate data cluster.

Appendix A.2 shows a formal definition of the above "merger" and "merger cost" process. Appendix A.3 shows the details of algorithm and its computational complexity.

# 4. Experiments

## 4.1. Experiments Setup

Eight multi-modal datasets are used, including 20NG, COIL20, Event, Soccer, 17Flowers, 75Flowers, COIL100 and MMI. The brief information of them is summarized in Table 1 and their details are shown in the Appendix C.1.

We compare with 4 traditional single-modal clustering methods, including KM, Ncuts, KM-ALL, Ncuts-ALL, and 13 state-of-the-art multi-modal clustering methods, including MVIB, Co(reg), MfIB, RMSC, LMSC, MLAN, GMC, DMIB, FPMVS-CAG, MCMLE, TBGL, TIM and SMVAGC-SF. Their details are shown in the Appendix C.2.

For all the compared methods, we adopt the parameter value settings from their papers to attain the best clustering results with optimal parameter setting on each dataset. For the proposed method, we search the parameters from the settings in the following parameter analysis section. Afterwards, we conduct the experiments for 10 times and report the average results and standard deviation in terms of Accuracy (Acc) and Normalized Mutual Information (NMI). All the compared methods and the proposed method are conducted in the same experimental environment, which is a desktop computer with Windows 10 operating system, 32GB RAM, and MATLAB 2021a.

Due to the limited space, the T-SNE visualization analysis is shown in Appendix C.3.

*Table 1.* Brief Information of the Datasets

| Dataset | Type | # Modality | # Samples | # Clusters |
|---------|------|-----------|-----------|-----------|
| 20NG | Text | 3 | 500 | 5 |
| COIL20 | Image | 3 | 1440 | 20 |
| Event | Image | 3 | 1579 | 8 |
| Soccer | Image | 3 | 280 | 7 |
| 17Flowers | Image | 3 | 1360 | 17 |
| 75Flowers | Image | 2 | 5514 | 75 |
| COIL100 | Image | 2 | 7200 | 100 |
| MMI | Video | 2 | 1760 | 22 |

## 4.2. Results and Analysis

We present the comparison clustering results with state-of-the-art methods on 8 multi-modal datasets in terms of Acc and NMI on Table 2 and 3. Observed from both the tables, we have the following discoveries.

**Single-modal VS All-modal.** Intuitively, concatenating the features from different modalities may improve the clustering quality in comparison to the single-modal clustering. However, from both tables, the clustering results degrade significantly on the multi-modal COIL20 and Soccer datasets. For instance, compared to Ncuts method on COIL20 dataset, the Acc and NMI values of Ncuts-All decrease by 28.55% and 26.08%, respectively. This clearly shows the instability of the all-modal methods and also reveals the necessity of multi-modal clustering.

**Single/all-modal VS Compared multi-modal.** Overall, the compared MMCs can beat the single/all-modal methods on most multi-modal datasets. Additionally, the second best results are always reached by the compared multi-modal method for all the datasets. However, it is also seen that on some datasets the MMCs is inferior to the single/all-modal methods. This is probably because that some compared MMCs tend to be trapped into local optimal solution and thus lead to unsatisfactory results.

**Single/all-modal VS Ours.** The proposed method outperforms the single/all-modal methods by a large margin on all the involved multi-modal datasets. For a notable example, our PTIB obtains an improvement of 57% and 42.07% in terms of Acc and NMI compared to the best values of single/all-modal methods on 20NG dataset. This phenomenon clearly demonstrates the superiority of the proposed method.

**Compared multi-modal VS Ours.** For the compared multi-modal IB-based methods, they have reached the second best results by two times on 17 flowers and MMI datasets. This mainly lies in the fact that the compared IB-based methods have the strong ability of the mutual information on the variable correlation quantization, while the remaining compared methods fail to do that. The compared multi-modal non-IB-based methods achieve the most second best clustering

*Table 2.* Clustering Results (+/-Standard Deviation) on First Four Datasets with SOTA Methods (● denotes the best result, ○ denotes the second best. )

| Method | 20NG | | COIL20 | | Event | | Soccer | |
|---|---|---|---|---|---|---|---|---|
| | Acc | NMI | Acc | NMI | Acc | NMI | Acc | NMI |
| KM | 22.28±1.48 | 4.45±2.32 | 53.06±3.20 | 65.06±2.12 | 33.93±4.10 | 19.84±2.69 | 25.82±5.06 | 18.70±7.97 |
| Ncuts (TPAMI'00) | 42.80±2.40 | 27.65±2.01 | 74.69±1.30 | 84.01±0.54 | 34.10±1.28 | 14.97±0.40 | 48.21±1.14 | 45.02±2.21 |
| KM-All | 21.46±0.68 | 1.76±0.65 | 46.14±6.58 | 60.70±4.51 | 28.85±2.29 | 11.37±2.10 | 22.46±3.94 | 8.14±3.59 |
| Ncuts-All (TPAMI'00) | 71.20±0.17 | 57.23±0.10 | 46.14±0.52 | 57.93±0.23 | 35.06±0.69 | 20.11±0.85 | 39.75±0.94 | 34.04±0.57 |
| MVIB (DASFAA'07) | 94.22±1.37 | 83.21±3.18 | 61.74±10.51 | 73.65±6.63 | 40.02±2.04 | 23.71±1.56 | 35.79±3.96 | 21.42±4.25 |
| Co(reg) (NeurIPS'11) | 20.02±0.62 | 3.15±0.54 | 64.33±1.68 | 83.79±0.45 | 38.58±0.92 | 24.30±0.55 | 24.13±0.53 | 11.43±0.39 |
| MfIB (IJCAI'13) | 93.76±2.89 | 85.11±4.54 | 83.81±4.29 | 92.39±1.97 | 48.58±1.50 | 33.41±1.35 | 53.64±2.76 | 49.74±3.44 |
| RMSC (AAAI'14) | 37.26±0.91 | 15.70±0.84 | 65.43±3.31 | 79.16±2.35 | 36.58±1.26 | 21.02±0.88 | 28.96±1.90 | 12.16±2.18 |
| LMSC (CVPR'17) | 96.16±0.57 | 88.37±1.54 | 71.94±2.72 | 82.18±2.37 | 43.92±2.84 | 27.53±2.58 | 31.25±6.53 | 15.85±8.71 |
| MLAN (TIP'18) | 96.40±0.11 | 89.18±0.17 | 87.22±2.30 ○ | 94.35±1.10 ○ | 19.90±0.72 | 6.66±0.80 | 28.21±0.01 | 21.27±0.17 |
| GMC (TKDE'20) | 98.20±0.00 | 93.92±0.00 | 60.90±0.00 | 84.67±0.00 | 18.11±0.00 | 10.74±0.00 | 29.29±0.00 | 25.82±0.00 |
| DMIB (TCYB'22) | 98.30±0.14 | 97.56±0.49 | 65.90±4.03 | 77.70±2.46 | 49.80±3.02 | 32.97±2.38 | 54.07±3.67 | 50.68±2.23 ○ |
| FPMVS-CAG (TIP'22) | 73.80±0.00 | 59.23±0.00 | 69.17±0.00 | 85.11±0.00 | 48.89±0.00 | 31.99±0.00 | 50.14±0.00 | 49.56±0.00 |
| MCMLE (TPAMI'22) | 77.40±0.00 | 69.96±0.00 | 85.83±0.00 | 93.48±0.00 | 44.46±0.00 | 30.24±0.00 | 56.07±0.00 ○ | 50.06±0.00 |
| TBGL (TPAMI'23) | 89.11±0.00 | 83.45±0.00 | 86.10±0.00 | 92.41±0.00 | 42.84±0.00 | 28.40±0.00 | 54.39±0.00 | 49.78±0.00 |
| TIM (TIP'23) | 99.40±0.00 ○ | 98.08±0.00 ○ | 56.70±4.08 | 71.39±0.29 | 54.60±2.50 | 36.86±1.75 | 48.93±0.51 | 41.42±4.09 |
| SMVAGC-SF (TIP'24) | 86.07±6.40 | 72.61±3.59 | 75.66±5.10 | 89.43±2.11 | 54.76±1.27○ | 36.97±0.65○ | 45.14±1.56 | 29.61±1.85 |
| PTIB | **99.80±0.00 ●** | **99.30±0.00 ●** | **93.33±0.00 ●** | **96.46±0.00 ●** | **60.24±0.16 ●** | **45.36±0.28 ●** | **62.86±0.17 ●** | **53.23±0.16 ●** |
| Improve (● VS ○) | 0.40 (↑) | 1.22 (↑) | 6.11 (↑) | 2.11 (↑) | 5.48 (↑) | 8.39 (↑) | 6.79 (↑) | 2.55 (↑) |

*Table 3.* Clustering Results (+/-Standard Deviation) on Remaining Four Datasets with SOTA Methods (● denotes the best result, ○ denotes the second best.)

| Method | 17Flowers | | 75Flowers | | COIL100 | | MMI | |
|---|---|---|---|---|---|---|---|---|
| | Acc | NMI | Acc | NMI | Acc | NMI | Acc | NMI |
| KM | 22.41±1.67 | 24.31±1.14 | 19.48±0.85 | 35.21±0.75 | 27.96±1.78 | 58.13±1.52 | 26.89±2.95 | 44.15±1.60 |
| Ncuts (TPAMI'00) | 27.71±0.72 | 26.43±0.40 | 24.80±0.58 | 41.50±0.19 | 40.97±1.28 | 58.52±0.59 | 38.43±0.47 | 53.17±0.43 |
| KM-All | 17.63±1.27 | 13.55±1.86 | 21.13±0.88 | 32.57±0.71 | 29.25±1.57 | 50.55±2.15 | 27.11±1.81 | 38.76±1.59 |
| Ncuts-All (TPAMI'00) | 28.77±0.63 | 26.31±0.27 | 27.41±0.31 | 42.41±0.21 | 48.63±0.97 | 64.74±0.56 | 40.53±1.52 | 52.77±0.62 |
| MVIB (DASFAA'07) | 21.32±1.05 | 18.28±1.48 | 18.49±0.61 | 33.05±0.45 | 46.71±2.30 | 70.29±1.10 | 44.95±2.60 ○ | 54.65±1.49 |
| Co(reg) (NeurIPS'11) | 26.28±0.49 | 27.12±0.20 | 28.16±0.36 | 44.95±0.09 | 48.35±0.44 | 70.86±0.15 | 34.72±0.53 | 51.31±0.22 |
| MfIB (IJCAI'13) | 38.52±2.03 | 37.24±1.40 ○ | 24.57±0.32 | 40.79±0.37 | 50.52±0.08 | 72.81±0.46 | 40.14±2.09 | 52.50±1.69 |
| RMSC (AAAI'14) | 19.70±0.66 | 17.86±0.38 | 26.42±0.97 | 42.95±0.30 | 46.32±0.28 | 69.33±0.45 | 30.28±1.05 | 43.94±0.89 |
| LMSC (CVPR'17) | 33.29±2.29 | 31.49±1.60 | 24.58±0.90 | 42.50±0.59 | 48.76±1.45 | 66.74±0.85 | 40.17±1.88 | 51.62±1.29 |
| MLAN (TIP'18) | 24.32±1.91 | 22.21±1.24 | 25.58±0.53 | 34.16±1.15 | 45.05±0.41 | 59.55±0.53 | 38.15±0.05 | 52.68±0.04 |
| GMC (TKDE'20) | 6.76±0.00 | 4.78±0.00 | 18.52±0.00 | 30.96±0.00 | 38.86±0.00 | 67.55±0.00 | 35.60±0.00 | 55.65±0.00 ○ |
| DMIB (TCYB'22) | 35.48±6.04 | 32.56±5.47 | 26.72±1.13 | 43.13±0.79 | 50.33±1.88 | 72.57±0.87 | 41.10±2.65 | 52.96±2.10 |
| FPMVS-CAG (TIP'22) | 30.51±0.00 | 27.27±0.00 | 23.83±0.00 | 38.24±0.00 | 45.03±0.00 | 70.58±0.00 | 36.77±0.00 | 51.03±0.00 |
| MCMLE (TPAMI'22) | 32.13±0.00 | 32.11±0.00 | 28.76±0.00 | 47.03±0.00 | 50.47±0.00 | 74.59±0.00 | 42.04±0.00 | 52.97±0.00 |
| TBGL (TPAMI'23) | 31.07±0.00 | 32.46±0.00 | 26.52±0.00 | 47.09±0.00 ○ | 51.66±0.00 | 67.82±0.00 | 43.15±0.00 | 53.27±0.00 |
| TIM (TIP'23) | 32.98±3.28 | 29.36±3.60 | 21.83±0.60 | 26.23±1.24 | 51.43±1.72 | 74.98±0.70 | 28.98±1.57 | 39.56±2.96 |
| SMVAGC-SF (TIP'24) | 42.41±2.07○ | 36.40±1.43 | 31.92±0.63○ | 46.89±0.22 | 56.78±1.93○ | 76.78±0.55○ | 40.93±2.14 | 53.03±1.25 |
| PTIB | **45.29±0.05 ●** | **42.49±0.08 ●** | **35.73±0.36 ●** | **51.91±0.20 ●** | **61.17±0.23 ●** | **82.86±0.19 ●** | **48.30±0.80 ●** | **60.39±0.49 ●** |
| Improve (● VS ○) | 2.88 (↑) | 5.25 (↑) | 3.81 (↑) | 4.82 (↑) | 4.39 (↑) | 6.08 (↑) | 3.35 (↑) | 4.74 (↑) |

results, but still underperform the proposed method.

**Improvement Analysis.** Compared with the TIM, MLAN, SMVAGC-SF, MCMLE and MVIB that obtained the second best results, the proposed method has attained notable improvements about 0.4%, 6.11%, 5.48%, 6.79%, 2.88%, 3.81%, 4.39% and 3.35% on the eight multi-modal datasets in terms of Acc values respectively. For weighted MMCs, the trustworthiness of the learned modality weights is significant for ensuring better clustering performance. The proposed method explicitly considers this vital factor by jointly incorporating the peer-review and trustworthy score.

### 4.3. Parameter Analysis

There is only one parameter $\beta$ in the proposed PTIB method, where we give the parameter settings as $[10, 50, 100, 500, 700, 900, 1000]$. Then we conduct extensive experiments on all the datasets with different settings to investigate the parameter sensitivity, and show the clustering performance with both Acc and NMI results in Figure 7.

From the figure, we can clearly obtain the best parameter settings corresponding to the optimal clustering results for different datasets, which are $[10, 50, 100, 500, 700, 900, 1000]$, $[500, 700, 900, 1000]$, $[50]$, $[50, 100]$, $[700, 900, 1000]$,

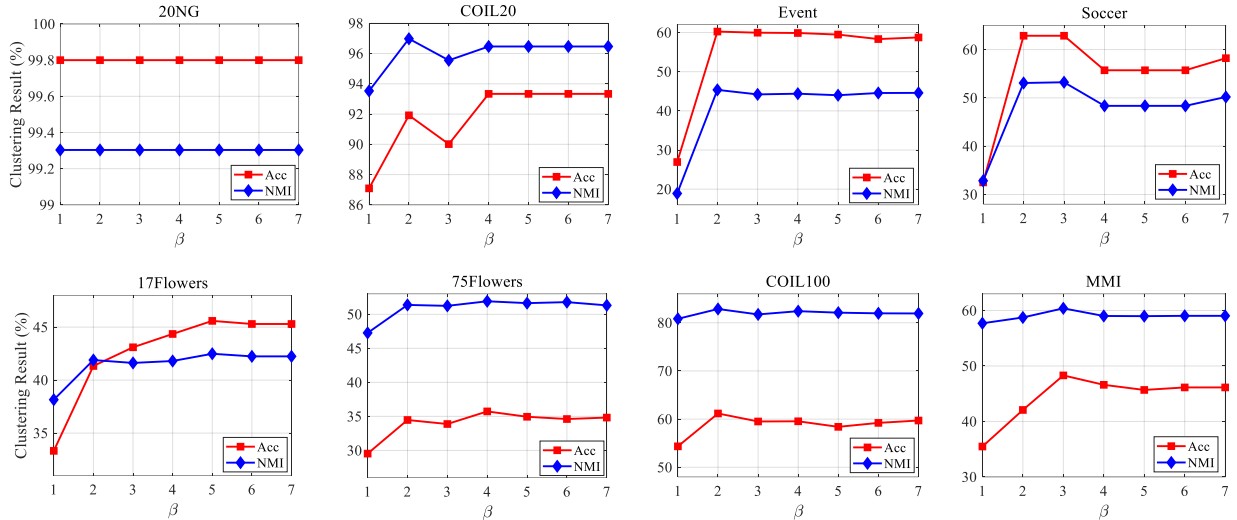

*Figure 7.* Parameter study of our PTIB method on eight multi-modal datasets. Note that the values in the horizontal axis indicate the seven parameters $[10, 50, 100, 500, 700, 900, 1000]$.

*Table 4.* Clustering Results (+/-Standard Deviation) on multi-modal Datasets with Parameter-free Version

| Datasets | PTIB | | Parameter-free PTIB | | Versus Margin | |
|---|---|---|---|---|---|---|
| | Acc | NMI | Acc | NMI | Acc | NMI |
| 20NG | 99.80±0.00 | 99.30±0.00 | 99.80±0.00 | 99.30±0.01 | 0.00 | 0.00 |
| COIL20 | 93.33±0.00 | 96.46±0.00 | 86.46±0.00 | 93.80±0.00 | -6.87 | -2.66 |
| Event | 60.24±0.16 | 45.36±0.28 | 59.01±0.62 | 44.39±0.50 | -1.23 | -0.97 |
| Soccer | 62.86±0.17 | 53.23±0.16 | 59.64±0.00 | 51.65±0.01 | -3.22 | -1.58 |
| 17Flowers | 45.29±0.05 | 42.49±0.08 | 42.74±1.38 | 40.92±0.82 | -2.55 | -1.57 |
| 75Flowers | 35.73±0.36 | 51.91±0.20 | 34.57±0.36 | 51.23±0.24 | -1.16 | -0.68 |
| COIL100 | 61.17±0.23 | 82.86±0.19 | 59.93±0.61 | 82.24±0.30 | -1.24 | -0.62 |
| MMI | 48.30±0.80 | 60.39±0.49 | 44.26±0.01 | 58.38±0.00 | -4.04 | -2.01 |

$[500]$, $[50]$, and $[100]$, respectively. It is observed that there are more than one best parameters for most datasets, such as 20NG, COIL20, and 17Flowers. Additionally, for all the datasets, the better results are obtained approximately from the range $[500, 700, 900, 1000]$, which is a wide range for practical parameter selection. All these observations reveal that the parameter tuning in practice is not a heavy burden and also shows the great potential of the proposed method for real-world applications.

### 4.4. Potential for Parameter-free Version

To further investigate the potential of the proposed method in practical applications, we make parameter $\beta$ as $+\infty$, leading to an elegant parameter-free version of our PTIB, formulated as $\mathcal{F}_{max}[p(t|x)] = \sum_{i=1}^{m} w^i \cdot I(T; Y^i)$. Then we conduct a series of experiments on involved eight multi-modal datasets, as shown in Table 4.

We also add a new column called versus margin, using the value of parameter-free version minus that of the PTIB method to compare the difference between them. From the table, we observe that the versus margin value always reaches roughly $-2\%$, and leads to about $-4\%$ on only few datasets (e.g., obtaining $-6.87\%$ on COIL20 datasets in terms of Acc value). This phenomenon demonstrates that the proposed method has a huge potential for more real-world applications, especially in the fields where we hardly tune the parameters without any prior knowledge.

## 5. Conclusion

Motivating by the peer-review mechanism in academia, in this article we propose a novel peer-review trustworthy information bottleneck (PTIB) method for addressing weighted multi-modal clustering problem. PTIB measures the modality correlations by iteratively learning the peer-review score of each modality, which works by alternately playing the role of "author" and "reviewer" for each modality. Additionally, we further design a trustworthy score for improving the reliability of the learned peer-review score, thus leading to more accurate modalities weights. Rich experiments on 8 datasets reveal the superiority and effectiveness of the proposed PTIB. The proposed method is also with some possible weaknesses. It is designed for fully aligned multi-modal clustering and complete multi-modal clustering, where none of the data samples across modalities are unaligned, missing, or damaged. And it requires the number of clusters of the dataset in advance, like almost all the existing multi-modal clustering methods. In future, we will focus on solving the discussed possible weaknesses of PTIB and apply it into more real-world applications. Moreover, this "peer-review" idea may be considered using for other problems, such as federated learning.

## Acknowledgements

The authors thank anonymous reviewers for their constructive comments. This work was supported by National Natural Science Foundation of China under Grant 62206254, Henan Province Outstanding Youth Science Fund Program under Grant 252300421223 and China Postdoctoral Science Foundation under Grant 2024T170843 and 2023M743186.

## Impact Statement

This paper presents work whose goal is to advance the field of Machine Learning. There are many potential societal consequences of our work, none which we feel must be specifically highlighted here.

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

# A. Theorem Proof and Optimization

## A.1. Proof of Theorem 3.3

*Proof.* Given a dataset with its true category of $\theta$ clusters and a specific clustering result, for a certain cluster, the probability of its major category is denoted as $a$ and the sum of the probabilities of minor categories is $1 - a$, leading to the first probability distribution $\{a, 1 - a\}$. It is assumed that the minor categories obey an arbitrary probability distribution $B = \{b_1, b_2, \ldots, b_{\theta-1}\}$, i.e., $\sum_j b_j = 1$, leading to the second probability distribution $\{a, b_1(1-a), b_2(1-a), \ldots, b_{\theta-1}(1-a)\}$. Then, we use the entropy of the second distribution to minus that of the first distribution as follows

$$
H_2(\{a, b_1(1-a), b_2(1-a), \ldots, b_{\theta-1}(1-a)\}) - H_1(\{a, 1-a\})
$$

$$
= -a\log_2 a - \sum_{j=1}^{\theta-1}[b_j(1-a)]\log_2[b_j(1-a)] + a\log_2 a + (1-a)\log_2(1-a)
$$

$$
=(1-a)\log_2(1-a) - \sum_{j=1}^{\theta-1}[b_j(1-a)][\log_2 b_j + \log_2(1-a)]
$$

$$
=(1-a)\log_2(1-a) - [\sum_{j=1}^{\theta-1}(1-a)b_j\log_2 b_j + \sum_{j=1}^{\theta-1}b_j(1-a)\log_2(1-a)] \tag{14}
$$

$$
=(1-a)\log_2(1-a) - (1-a)\sum_{j=1}^{\theta-1}b_j\log_2 b_j - (1-a)\log_2(1-a)
$$

$$
=(1-a)[-\sum_{j=1}^{\theta-1}b_j\log_2 b_j]
$$

$$
=(1-a)H(B)
$$

Since $1 - a \geq 0$ and $0 \leq H(B) \leq log_2(\theta - 1)$, we have $0 \leq H_2 - H_1 \leq (1-a)log_2(\theta - 1)$, which clearly shows that, the more mixed the minor categories are, the higher the entropy of it gets, thus resulting in high uncertainty of the cluster. Actually, $H(B)$ is the noisy and uncertain information that the minor categories convey, which interferes with the judgment of useful information.

This proves the theorem. □

## A.2. The Formal Definition of the Optimization Process

**Proposition A.1.** *[Merger] (Hu et al., 2020) If the separate cluster $\{x\}$ is merged into one specific cluster $t$ in the $i$-th modality, a new merged cluster is reached, named $\hat{t}$. This process is formulated as the following:*

$$
\begin{cases}
p(\hat{t}) = p(x) + p(t) \\
p(y^i|\hat{t}) = \pi_1 \cdot p(y^i|x) + \pi_2 \cdot p(y^i|t)
\end{cases} \tag{15}
$$

*where $p(y^i|x)$ indicates the feature conditional distribution of the $i$-th modality, $p(y^i|t)$ indicates the cluster centroid of the $i$-th modality, and the merger function $\Pi = \{\pi_1, \pi_2\} = \{\frac{p(x)}{p(\hat{t})}, \frac{p(t)}{p(\hat{t})}\}$.*

*Proof.* The detailed proof can be referred from the Proposition 1 in the work (Hu et al., 2020). □

To obtain the maximal value of the function $Eq.(13)$, we attempt to select the optimal cluster $t^{new}$ for each merger, and ensure the "merger cost" i.e., the value change of function $Eq.(13)$, is always minimal. This is formulated by

$$
t^{new} = \arg\min(\Delta\mathcal{F}_{max}) = \arg\min(\mathcal{F}_{max}^{bef} - \mathcal{F}_{max}^{aft}), \tag{16}
$$

where $\mathcal{F}_{max}^{bef}$ and $\mathcal{F}_{max}^{aft}$ indicate the value of $Eq.(13)$ before and after the merger process respectively.

Actually, the above formulation has a significant impact on whether a good or bad new cluster $t^{new}$ is selected. We thus have the merger cost formulation with the following theorem.

---

**Algorithm 1** The Proposed PTIB

---

1: **Input:** $m$ joint distributions $\{p(X, Y^i)\}_{i=1}^m$, the number of clusters $|T|$, the balance parameter $\beta$.
2: **Output:** Final clustering result $p(t|x)$.
3: **Modality Weight Initialization:** Compute the initial modality weights with initial peer-review and trustworthy score;
4: **Random Clustering:** $T \leftarrow$ Random partition of $\mathcal{X}$ into $|T|$ clusters;
5: **repeat**
6:     **for all** $x \in \mathcal{X}$ **do**
7:         **Draw:** Draw $x$ from the "old" cluster $t^{old}$ to become a separate cluster $\{x\}$;
8:         **Merger:** Select a "new" cluster $t^{new}$ for the separate cluster $\{x\}$ to merge corresponding to the minimal merger cost in Theorem A.2;
9:     **end for**
10:     Update the trustworthy score using the clustering result in each iteration;
11:     Update the weight for each modality;
12: **until** Samples in different clusters remain unchanged or a fixed number of iterations.

---

**Theorem A.2.** *[Merger Cost] (Hu et al., 2020) Given two clusters $\{x\}$ and $t$, we have the merger cost as*

$$\Delta \mathcal{F}_{max}(\{x\}, t) = p(\hat{t}) \cdot dist(\{x\}, t), \tag{17}$$

*where*

$$dist(\{x\}, t) = \sum_{i=1}^m w^i [JS_\Pi[p(y^i|x), p(y^i|t)] - \beta^{-1} JS_\Pi[p(y(x), p(x|t)]].$$

*where $JS$ is the* Jensen-Shannon *divergence (Lin, 2006).*

*Proof.* The detailed proof can be referred from the Theorem 1 in the work (Hu et al., 2020). $\square$

### A.3. Algorithm and Computational Complexity

The Algorithm 1 shows the details of the optimization process and we investigate the computational complexity of it. At step 8 in Algorithm 1, given sample $x$, we have the merger cost as $\Delta \mathcal{F}_{max}(\{x\}, t)$ for every "new" cluster $t$ to reach the minimal one, and it takes $O(|T||X|(|Y^1| + |Y^2| \cdots + |Y^m|))$. When the samples in different clusters remain unchanged or after a fixed number of iterations, it takes $O(r|T||X|(|Y^1| + |Y^2| \cdots + |Y^m|))$, where $r$ is the number of repetitions. Generally, the number of clusters is treated as constant. Hence, the overall computational complexity takes $O(r|X|(|Y^1| + |Y^2| \cdots + |Y^m|))$.

## B. Discussion

In this section, we first discuss and show the differences or advantages of the proposed method with existing multi-modal clustering methods. Then, we further analyze the weaknesses of our method.

The main strengths of the PTIB method in comparison with existing methods are as follows.

- **Trustworthy weight learning.** To our knowledge, none of the existing weighted MMCs employ the trustworthy strategy in the weight learning process, which may probably leads to inaccurate modality weights. Unlike them, we attempt to learn trustworthy modality weights in an iterative optimization process.

- **Correlation quantization based learning.** We in this paper focus on modality correlation quantization using mutual information, while most related existing methods generally measure themselves for weight learning, e.g., using objective function value of each modality for complementary information learning.

- **Parameter-free weight learning.** Most weighted MMCs usually adopt one or more regularization parameters to control the learned weight distribution, where the parameters are difficult to tune in practice. In contrast, the weight learning process in this paper is completely parameter-free without need any prior knowledge.

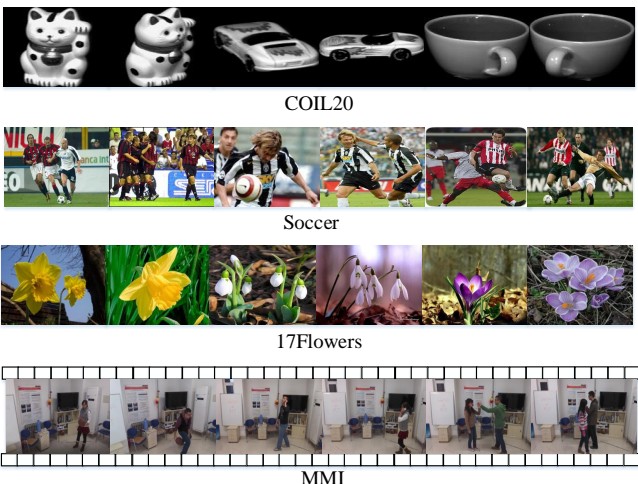

*Figure 8.* Some typical images and videos from COIL20, Soccer, 17Flowers and MMI datasets.

- **Self-supervision mechanism.** Self-supervised learning mechanism (Xu et al., 2023) is incorporated into the proposed method for guiding the modality weight learning. Thus, in this way both the clustering structure learning and the weight learning can mutually benefit from win-win cooperation.

Some possible weaknesses of the proposed method are revealed in the following.

- **Fully aligned multi-modal clustering.** The proposed method works under the assumption that each data sample across different modalities is fully aligned. For example of the multi-feature images, the first sample described with one kind of feature, e.g., shape, must be aligned with the same sample described with another feature, e.g., color.

- **Complete multi-modal clustering.** The proposed method can only solve the complete multi-modal clustering problem where none of the data samples across modalities are missing or damaged. Incomplete multi-modal clustering (Wen et al., 2023; 2024) or its self-supervised version (Huang et al., 2023) has attracted lots of attention and is worth considering in the future.

- **Given number of clusters in advance.** Like almost all the existing multi-modal clustering methods, the proposed method requires the number of clusters of the dataset in advance, which may limit its wide applications in completely unknown areas.

## C. More Experimental Details

### C.1. The Detail of Datasets

Here, we provide detailed description of the dataset used in the experimental part of this paper and some exemplar samples from the image and video data are shown in Figure 8.

**20NG dataset** [1] is composed of 500 newsgroup documents extracted from the 20 Newsgroup dataset. Three different pre-processing methods provide the modalities for each document.

**COIL20 dataset** [2] has 1440 images about 20 objects, where each object has 72 images taken at $5°$ intervals in its $360°$ horizontal rotation. Three features are adopted for shape, color and texture representation, i.e., SIFT (Lowe, 2004), Color Attention (Khan et al., 2009), and TPLBP (Wolf et al., 2008) respectively. Every feature represents one single modality.

**Event dataset** [3] contains eight kinds of sports event classes with 1579 images. The features are the same as those of the COIL20 dataset.

---

[1] http://lig-membres.imag.fr/grimal/data.html
[2] http://www.cs.columbia.edu/CAVE/software/softlib/coil-20.php
[3] http://vision.stanford.edu/lijiali/event_dataset/

**Soccer dataset** [4] contains 280 images of 7 soccer teams captured from the websites. The features extracted by three ways, including SIFT (Lowe, 2004), Color Attention (Khan et al., 2009), and TPLBP (Wolf et al., 2008), are used as three modalities.

**17Flowers dataset** [5] consists of 1360 images of flowers belonging to 17 different classes, each of which has 80 images. Three kinds of features, i.e., SURF (Bay et al., 2006), Color Attention (Khan et al., 2009), and TPLBP (Wolf et al., 2008), are used as three modalities. This dataset is challenging for clustering because there are classes with large variations within each class and close similarity across classes.

**75Flowers dataset** [6] is selected from the 102 Flowers dataset. The images contain a large variation on the posture and the light. This dataset contains two modalities by using SIFT (Lowe, 2004) and Color Attention (Khan et al., 2009) as shape and color feature extractors respectively.

**COIL100 dataset** [7] consists of images from 100 objects, where each object has 72 images. We use two features extracted by SIFT (Lowe, 2004) and SURF (Bay et al., 2006) as two modalities.

**MMI dataset** [8] has 1760 samples with challenging 22 multi-modal (RGB, depth and Skeleton) and multi-modal (front and side modalities) interactive human actions, which are collected in cluttered and unclear places. Note that we adopt the RGB action videos with two modalities for experiments.

## C.2. The Detail of Comparison Methods

1. **KM, Ncuts** (Shi & Malik, 2000) : KM ($k$-means) and Ncuts are traditional single-modal clustering methods, and the best results are reported among different modalities for each dataset.

2. **KM-All, Ncuts-All**: Both of the methods are built by applying their single-modal version on the multi-modal datasets with concatenated features.

3. **MVIB** (Gao et al., 2007): It is the first multi-view IB method proposed to address the document clustering problems from the websites, working by designing a compatible constraint to ensure the consistency among view assignments.

4. **Co(reg)** (Kumar et al., 2011): It co-regularizes the data clustering hypotheses among views to learn consistent assignments based on spectral model.

5. **MfIB** (Lou et al., 2013): It is a weighted multi-feature IB method designed for solving the unsupervised image classification, where the weights are set manually.

6. **RMSC** (Xia et al., 2014): It solves the noisy multi-view clustering problem by designing a robust spectral method.

7. **LMSC** (Zhang et al., 2017): It learns latent shared representations among views to make the feature subspace more robust and accurate.

8. **MLAN** (Nie et al., 2018): It jointly learns the local structure and clustering assignments, and then automatically tunes the view weights without using parameters.

9. **GMC** (Wang et al., 2020a): It is a graph-based weighted multi-view clustering method by automatically tuning the algorithm parameters.

10. **DMIB** (Hu et al., 2022): It jointly takes account into the dual correlations about the cross-feature and cross-cluster view correlations for multi-view clustering based on IB theory.

11. **FPMVS-CAG** (Wang et al., 2022): It deals with the multi-view subspace clustering problem by a fast parameter-free method with the guidance of selected consensus anchors.

12. **MCMLE** (Zhong & Pun, 2022): It improves the traditional Ncuts method for multi-view clustering by Laplacian embedding to learn a shared binary assignment matrix among different modalities.

---

[4] http://lear.inrialpes.fr/people/vandeweijer/data.html
[5] http://www.robots.ox.ac.uk/~vgg/data/flowers/17/index.html
[6] http://www.robots.ox.ac.uk/~vgg/data/flowers/102/index.html
[7] http://www.cs.columbia.edu/CAVE/software/softlib/coil-100.php
[8] http://media.tju.edu.cn/m2i.html

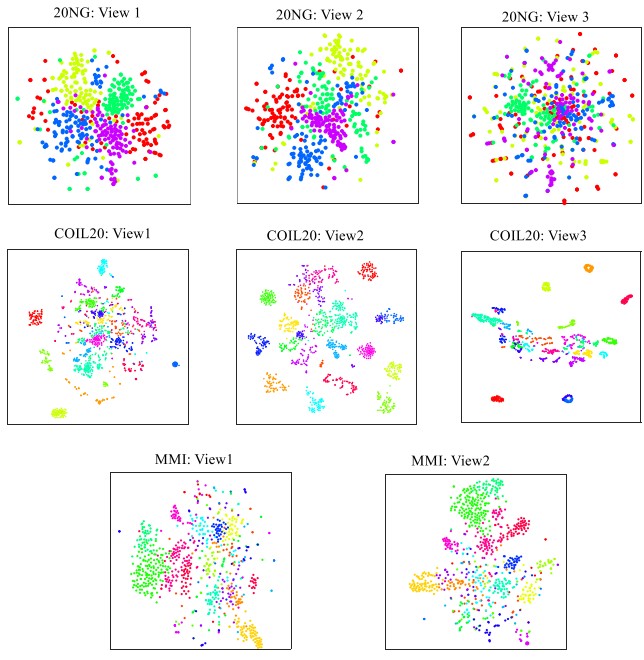

*Figure 9.* T-sne visualization results on 20NG, COIL20 and MMI datasets.

13. **TBGL** (Xia et al., 2023a): It focuses on learning tensorized bipartite graphs for clustering multi-view datasets by simultaneously considering the intra/inter-view similarities.

14. **TIM** (Zhang et al., 2023): It is an information-theoretical method for solving the multi-view clustering problem, and works by following three principles, i.e., contained, complementary and compatible principle.

15. **SMVAGC-SF** (Wang et al., 2024): It jointly optimizes anchor graph construction and graph alignment, and adaptively fuses multiple anchor graphs with different magnitudes to improve the quality of multi-view clustering.

## C.3. T-SNE Visualization Analysis

To further illustrate the learned clustering structure, we vividly show the $t$-SNE visualization of the clustering results in Figure 9 with three typical multi-modal datasets, i.e., 20NG, COIL20 and MMI. From this figure, it is observed that the visualization of most modalities from the involved datasets illustrate a relatively compact and separated clustering structure. For a typical example of the modality 2 and 3 in COIL20 dataset, the data clusters with different colors are quite clear, and data samples in most clusters are densely distributed while samples from different clusters are in a long distance.

