# OpenReview forum: "A Peer-review Look on Multi-modal Clustering: An Information Bottleneck Realization Method"
_ICML.cc/2025/Conference — ICML 2025 poster_

### Official Review · Reviewer_b4ga · 2025-03-08

**Overall Recommendation:** 4

**Summary:**

For the mentioned three limitations faced by most current weighted multimodal clustering methods, this paper, inspired by the peer-review mechanism in academia, iteratively considers one modality as the "author" and the remaining modalities as "reviewers" to obtain the peer-review score for each modality. To improve the reliability, a trustworthy score with a self-supervised working mechanism is further designed. Finally, a new PTIB method is proposed, and lots of experimental results show its effectiveness.

**Claims And Evidence:**

The claims made in the submission are supported by clear and convincing evidence. Lots of experimental results show its effectiveness over many compared methods.

**Essential References Not Discussed:**

There are no related works that are not currently discussed in the paper.

**Experimental Designs Or Analyses:**

I have checked the soundness/validity of the experimental designs or analyses, including subsection 4.1, 4.2, 4.3 and 4.4 and its corresponding analysis.

**Methods And Evaluation Criteria:**

The proposed method and evaluation criteria make sense for the problem. The proposed method iteratively considers one modality as the "author" and the remaining modalities as "reviewers" to obtain the peer-review score for each modality. Then, a trustworthy score with a self-supervised working mechanism is further designed to improve the reliability. The evaluation criteria such as datasets, evaluation metrics, compared methods in the experiments are frequently used in the community.

**Other Comments Or Suggestions:**

Please see the above, and I have no other comments. I am looking forward to the authors’ reply to my comments, which may help me to give a final rating.

**Other Strengths And Weaknesses:**

The authors are inspired by the peer-review mechanism in academia and propose a new and well-organized framework named PTIB method. The introduction, methods and experimental sections are articulated clearly. This paper is well written and the novelty is quite enough, and the problem is well addressed.

However, there are also some weaknesses below:
1. There are many trustworthy multi-modal classification or clustering methods published recent years. This paper also mentioned the 'trustworthy'. What are the differences between this paper and existing trustworthy multi-modal classification or clustering methods?
2. Methodological limitations are only mentioned in Appendix B and are not summarized in the conclusion section.
3. Lack of description of the experimental environment, which may help readers to better reproduce the main parts of the methods.
4. The font size in some pictures is small, as shown in Figure 2.

**Questions For Authors:**

Please see the above, and I have no other comments. I am looking forward to the authors’ reply to my comments, which may help me to give a final rating.

**Relation To Broader Scientific Literature:**

The proposed method is novel by integrating peer-review idea to address multi-modal clustering problem.

**Theoretical Claims:**

I have checked the correctness of the proofs for theoretical claims, including the theorem 3.3 and its proof in Appendix A.1.

---

> ### Author Rebuttal · Authors · 2025-03-31
>
> Thank you for the insightful comments and constructive suggestions. We have carefully revised the whole manuscript and provided detailed responses to each point below.
>
> **1. There are many trustworthy multi-modal classification or clustering methods published recent years. This paper also mentioned the 'trustworthy'. What are the differences between this paper and existing trustworthy multi-modal classification or clustering methods?**
>
> ***Response:*** Thanks for your comments. Regarding the trustworthiness of multiple modalities, almost all existing methods [1-4] focus on trustworthy multimodal classification. To ensure the reliability of both the multi-modal integration and the final decision, the trustworthy multimodal classification methods produce a stable and reasonable uncertainty estimation for each modality and thus promote both classification reliability and robustness. For example, Han et al. [2] introduce the variational Dirichlet to characterize the distribution of the class probabilities, parameterized with evidence from different views and integrated with the Dempster-Shafer theory, thus promoting both classification reliability and robustness. Zheng et al. [3] propose a trustworthy multimodal classification network via multi-level confidence learning which integrates both feature and label-level confidence learning for trustworthy multimodal classification. Zou et al. [4] induce a transparent fusion strategy based on the modality confidence estimation strategy to track information variation within different modalities for dynamical fusion.
>
> Different from them, the proposed method aims to guarantee the trustworthy of the learned modal weights in a self-supervised manner. To the best of our knowledge, none of the existing weighted MMCs employ the trustworthy strategy in the weight learning process.
>
> [1] Z. Han, C. Zhang, H. Fu, and J. T. Zhou, Trusted multi-view classification, in Proceedings of the International Conference on Learning Representations, 2021, pp. 1–11.
>
> [2] Han, Z., Zhang, C., Fu, H., and Zhou, J. T. Trusted multi-view classification with dynamic evidential fusion. IEEE transactions on pattern analysis and machine intelligence,2023, 45(2), 2551-2566.
>
> [3] Zheng, X., Tang, C., Wan, Z., Hu, C., and Zhang, W. Multi-level confidence learning for trustworthy multimodal classification. In Proceedings of the AAAI conference on artificial intelligence,2023, pp. 11381-11389.
>
> [4] Zou, X., Tang, C., Zheng, X., Li, Z., He, X., An, S., and Liu, X. Dpnet: Dynamic poly-attention network for trustworthy multi-modal classification. In Proceedings of the 31st ACM International Conference on multimedia, 2023, pp. 3550-3559.
>
> **2. Methodological limitations are only mentioned in Appendix B and are not summarized in the conclusion section.**
>
> ***Response:*** Thanks for your comments. We will summarize the limitations of the proposed method in the conclusion section:
> The proposed method is also with some possible weaknesses. It is designed for fully aligned multi-modal clustering and complete multi-modal clustering, where none of the data samples across modalities are unaligned and missing or damaged. And it requires the number of clusters of the dataset in advance like almost all the existing multi-modal clustering methods.
>
> **3. Lack of description of the experimental environment, which may help readers to better reproduce the main parts of the methods.**
>
> ***Response:*** Thanks for your comments. All the compared methods and the proposed method are conducted in the same experimental environment, which is a desktop computer with Windows 10 operating system, 32GB RAM, and MATLAB 2021a.
>
> **4. The font size in some pictures is small, as shown in Figure 2.**
>
> ***Response:*** Thanks for your comments. We will use a more legible font size for the figures in the revised version. And we checked all the figures in the manuscript to ensure that they were clear and readable enough.
>
> Thanks again for the valuable suggestions provided by the reviewer. The modifications will be added to the final version.

---

### Official Review · Reviewer_3BK8 · 2025-03-09

**Overall Recommendation:** 4

**Summary:**

In this paper, the authors propose a new multi-modal clustering method by information bottleneck method with an interesting peer-review look. This method work in a weighted mechanism with two learning scores, including peer-review and trustworthy score. It is noted that the weight learning process is conducted without parameter tuning, which is good for practical applications. Many experiments on benchmark datasets show the effectiveness and superiority of the proposed method.

## update after rebuttal

The authors have addressd my all concerns and I have reviewed the comments from the other three reviewers. So, I have raised my score.

**Claims And Evidence:**

The claims made in the submission can be supported by clear and convincing experiments, which provide different levels of validation on the proposed method.

**Essential References Not Discussed:**

Essential References are discussed in the paper.

**Experimental Designs Or Analyses:**

I have checked the soundness/validity of experimental designs or analyses in the submission. Many experiments on benchmark datasets show the effectiveness and superiority of the proposed method.

**Methods And Evaluation Criteria:**

The proposed methods and/or evaluation criteria make sense for the multi-modal clustering problem, especially the peer-review learning mechanism and many experimental results.

**Other Comments Or Suggestions:**

I have not any other comments or suggestions here. I have given all my comments in the “Other Strengths And Weaknesses”.

**Other Strengths And Weaknesses:**

Strengths
(1) novelty: novel for the ICML conference; notable advantages compared with state-of-the-art methods.
(2) soundness: technically sound, under a very rigorous framework.
(3) significance: the problem is significant and also the method may bring impact on the related community.

Weaknesses
(1) The peer-review look on multi-modal clustering is interesting, have the authors considered using this idea for other problems? The authors are encouraged to give some of them in the future work so that it may help the readers to explore more possibilities of this ideas into dealing with more problems.
(2) Some equations in the figure are too small, which may influence the readability.
(3) In section 2, it is not only to introduce IB method, some multi-modal clustering works using IB needs to be added here.

**Questions For Authors:**

I have not any other questions here. I have given all my comments in the “Other Strengths And Weaknesses”.

**Relation To Broader Scientific Literature:**

The key contributions are: the authors propose a new multi-modal clustering method by information bottleneck method with an interesting peer-review look. It is new to the area.

**Theoretical Claims:**

I have checked the correctness of proofs for theoretical claims in the submission, i.e., theorem 3.3 and its detailed proof.

---

> ### Author Rebuttal · Authors · 2025-03-31
>
> Thank you for the insightful comments and constructive suggestions. We have carefully revised the whole manuscript and provided detailed responses to each point below.
>
> **1. The peer-review look on multi-modal clustering is interesting, have the authors considered using this idea for other problems? The authors are encouraged to give some of them in the future work so that it may help the readers to explore more possibilities of this ideas into dealing with more problems.**
>
> ***Response:*** Thanks for your comments. It is possible to try to migrate the idea of "peer review" to federated learning. Federated Learning is a distributed machine learning technique. Its core concept is to enable multiple participants (e.g., devices or institutions) to collaboratively train a globally shared machine learning model without sharing local data, thereby achieving data privacy preservation and collaborative model building. A critical step in this process is secure aggregation: clients upload their locally trained model updates to a central server, which then merges these updates using aggregation algorithms (e.g., FedAvg aggregates local model updates from multiple clients via weighted averaging) to generate a new global model. In this “peer-review” framework, each client can be viewed as an "author" submitting its update, while the remaining clients act as "reviewers" to score the quality of that update. If a client's update has a low score, it may indicate that the client is a low-quality client or an external malicious node. By leveraging these scores, the server can reduce the importance of low-scoring updates or exclude low-scoring updates during aggregation, ensuring the robustness of the global model.
>
> **2. Some equations in the figure are too small, which may influence the readability.**
>
> ***Response:*** Thanks for your comments. We will use a more legible font size for the figures in the revised version. And we checked all the figures in the manuscript to ensure that they were clear and readable enough.
>
> **3. In section 2, it is not only to introduce IB method, some multi-modal clustering works using IB needs to be added here.**
>
> ***Response:*** Thanks for your comments. Some newly added multi-modal clustering works using IB are in the following:
> In recent years, IB theory has been widely used in various multi-modal clustering tasks. For example, Federici et al. [1] proposed a multi-modal IB method that can identify non-shared information between two modalities. Yan et al. [2] proposed a multi-modal IB method that uses shared representations of multiple modalities to eliminate private information of a single modality. But the modality-private information is eliminated as much as possible during the process of data compression, only exploring the shared information of modalities without taking advantage of the complex relationship between modalities. Hu et al. [3] conduct the information bottleneck theory in the origin data, the learned features of high-dimensional and the learned features of low-dimensional to fuse the module information. However, its final clustering result is obtained by directly averaging the local clusters from the modal high-dimensional features.
>
> Different from the existing multi-modal clustering methods based on IB theory, the proposed method considers the complex relationship between modalities, where the designed multi-modal peer review is used to reasonably score the contribution of each perspective, and the self-supervised trustworthy score is used to ensure the reliability of the process.
>
> [1] Federici, M., Dutta, A., Forre, P., Kushman, N., and Akata, Z. Learning robust representations via multi-view information bottleneck. arXiv preprint arXiv:2002.07017, 2020.
>
> [2] Yan, X., Mao, Y., Ye, Y., and Yu, H. Cross-modal clustering with deep correlated information bottleneck method. IEEE Transactions on Neural Networks and Learning Systems, 2023.
>
> [3] Hu, J., Yang, C., Huang, K., Wang, H., Peng, B., and Li, T. Information bottleneck fusion for deep multi-view clustering. Knowledge-Based Systems, 2024, 289, 111551.
>
> Thanks again for the valuable suggestions provided by the reviewer. The modifications will be added to the final version.

---

### Official Review · Reviewer_Qckw · 2025-03-10

**Overall Recommendation:** 4

**Summary:**

Most existing methods in multimodal clustering have three core challenges on the trustworthiness, weight learning and parameter learning. Motivated by the peer-review mechanism, this paper solves the multimodal clustering problem and realizes mutual review of different modalities by rotating the roles of "author" and "reviewer" to explore the potential relationship. Moreover, a trustworthy score is further given in a self-supervised manner. By considering the two aspects, a new peer-review trustworthy information bottleneck method is proposed. Comparative experiments on eight multimodal datasets show the proposed method outperforms existing state-of-the-art methods.

## update after rebuttal

The authors have addressed my most concerns and I keep my rating.

**Claims And Evidence:**

The claims made in the submission are supported by clear and convincing evidence, as demonstrated by the comparative experiments.

**Essential References Not Discussed:**

The introduction sufficiently covers essential related works, but additional discussion on the differences between multimodal and multi-view clustering would provide better context for its contributions. Some information theory based methods could also be reviewed, e.g., Dual Contrastive Prediction for Incomplete Multi-View Representation Learning, TPAMI 2023.

**Experimental Designs Or Analyses:**

The experimental design and analyses in Section 4 have been reviewed, and their soundness and validity are confirmed.

**Methods And Evaluation Criteria:**

After carefully checking the manuscript, the proposed methods and evaluation criteria make sense for the multi-modal clustering problem. The proposed peer-review trustworthy information bottleneck method is interesting, and comparative experiments on eight multimodal datasets show the proposed method outperforms existing state-of-the-art methods.

**Other Comments Or Suggestions:**

NA

**Other Strengths And Weaknesses:**

This paper proposes a novel multi-modal clustering method grounded in robust theory and practical applicability. The paper has a clear structure. The proposed method is validated with well-designed experiments.

The paper also has some limitations:
1.	Lack of discussion on existing trustworthy multimodal clustering approaches. Are there prior weighted multimodal clustering methods that incorporate trustworthiness in weight learning?
2.	Unclear details regarding implementation of baseline comparisons. Where was the code for the compared methods obtained? A fair comparison is necessary to validate the performance improvements.
3.	Many cited works focus on multi-view clustering, while this paper targets multimodal clustering. What are the key differences, and how do multimodal clustering methods differ from multi-view clustering approaches? The paper should include recent works on multimodal clustering for a more comprehensive discussion.

**Questions For Authors:**

See weaknesses.

**Relation To Broader Scientific Literature:**

The paper contributes to the broader literature by introducing a peer-review-based multimodal clustering framework, which has not been extensively explored in previous works. The trustworthiness score and information bottleneck approach further enhance its novelty.

**Theoretical Claims:**

The correctness of Theorem 3.3 and its proof has been reviewed and found to be valid.

---

> ### Author Rebuttal · Authors · 2025-04-01
>
> Thank you for the insightful comments and constructive suggestions. We have carefully revised the whole manuscript and provided detailed responses to each point below.
>
> **1. Lack of discussion on existing trustworthy multimodal clustering approaches. Are there prior weighted multimodal clustering methods that incorporate trustworthiness in weight learning?**
>
> ***Response：*** Thanks for your comments. Regarding the trustworthiness of multiple modalities, almost all existing methods [1-4] focus on trustworthy multimodal classification. To ensure the reliability of both the multi-modal integration and the final decision, the trustworthy multimodal classification methods produce a stable and reasonable uncertainty estimation for each modality and thus promote both classification reliability and robustness. For example, Han et al. [2] introduce the variational Dirichlet to characterize the distribution of the class probabilities, parameterized with evidence from different views and integrated with the Dempster-Shafer theory, thus promoting both classification reliability and robustness. Zheng et al. [3] propose a trustworthy multimodal classification network via multi-level confidence learning which integrates both feature and label-level confidence learning for trustworthy multimodal classification. Zou et al. [4] induce a transparent fusion strategy based on the modality confidence estimation strategy to track information variation within different modalities for dynamical fusion.
>
> Different from them, the proposed method aims to guarantee the trustworthy of the learned modal weights in a self-supervised manner. To the best of our knowledge, none of the existing weighted MMCs employ the trustworthy strategy in the weight learning process.
>
> [1] Z. Han, C. Zhang, H. Fu, and J. T. Zhou, Trusted multi-view classification, in Proceedings of the International Conference on Learning Representations, 2021, pp. 1–11.
>
> [2] Han, Z., Zhang, C., Fu, H., and Zhou, J. T. Trusted multi-view classification with dynamic evidential fusion. IEEE transactions on pattern analysis and machine intelligence,2023, 45(2), 2551-2566.
>
> [3] Zheng, X., Tang, C., Wan, Z., Hu, C., and Zhang, W. Multi-level confidence learning for trustworthy multimodal classification. In Proceedings of the AAAI conference on artificial intelligence,2023, pp. 11381-11389.
>
> [4] Zou, X., Tang, C., Zheng, X., Li, Z., He, X., An, S., and Liu, X. Dpnet: Dynamic poly-attention network for trustworthy multi-modal classification. In Proceedings of the 31st ACM International Conference on multimedia, 2023, pp. 3550-3559.
>
> **2. Unclear details regarding implementation of baseline comparisons. Where was the code for the compared methods obtained? A fair comparison is necessary to validate the performance improvements.**
>
> ***Response：*** Thanks for your comments. The codes of the all compared methods are obtained from the GitHub websites published in the original papers, and their parameter settings also strictly follow the implementation details in the original papers to ensure correct reproduction. All the compared methods and the proposed method are conducted in the same experimental environment, which is a desktop computer with Windows 10 operating system, 32GB RAM, and MATLAB 2021a.
>
> **3. Many cited works focus on multi-view clustering, while this paper targets multimodal clustering. What are the key differences, and how do multimodal clustering methods differ from multi-view clustering approaches? The paper should include recent works on multimodal clustering for a more comprehensive discussion.**
>
> ***Response：*** Generally, the aim of multi-view clustering and multimodal clustering is similar especially in integrating different sources of information for improving clustering performance. The differences between them are as follows: Multi-view learning focuses on diverse feature representations of the same object, while multi-modal learning deals with complex relationships between heterogeneous modalities, which is more complicated to handle. In practical applications, overlaps between them may exist (e.g., multi-modal data can also be considered as generalized multi-view data). However, technical solutions should be selected based on data characteristics (feature homogeneity and semantic consistency) to ensure methodological compatibility. In this paper, we use a more general expression of multi-modal clustering instead of multi-view clustering. In the final version, we will include and discuss more recent works on multi-modal clustering. Moreover, we will also discuss the relationships and differences between them as well as some information theory based methods, e.g., 'Dual Contrastive Prediction for Incomplete Multi-View Representation Learning', in the final version.
>
> Thanks again for the valuable suggestions provided by the reviewer. The modifications will be added to the final version.

---

### Official Review · Reviewer_ZBoD · 2025-03-13

**Overall Recommendation:** 4

**Summary:**

This paper proposes a new peer-review trustworthy information bottleneck method. It designs a multimodal peer review process, in which the modality will iteratively act as an "author" or "reviewer" to conduct peer review to explore the potential relationship, which is quantified as the peer-review score; and the trustworthiness of the modality as a "reviewer" is judged in a self-supervised manner. Extensive experiments on 8 datasets show that the proposed method is superior to existing cutting-edge methods in terms of clustering performance indicators.

**Claims And Evidence:**

The claims in the manuscript are supported by the clear and convincing evidence from the extensive experiments and the results from different aspects.

**Essential References Not Discussed:**

None

**Experimental Designs Or Analyses:**

I have checked the soundness/validity of the experimental designs or analyses from the Sec 4.

**Methods And Evaluation Criteria:**

The proposed methods and/or evaluation criteria (including the datasets, compared methods, performance indicators) make sense for the problem.

**Other Comments Or Suggestions:**

Refer to the weakness

**Other Strengths And Weaknesses:**

The authors provide a clear explanation of the proposed method, enhancing the reader's understanding of its importance. The paper presents a clear motivation and is well-structured, which also offers some valuable insights. My comments are shown as: First, generally, in a peer-review process, an EiC is also involved. I am interesting about which part stands for the EiC role in the clustering process. Second, the author should detail the ‘k-means-like draw-and-merger algorithm’ in Sec 3.5 using one or two sentences to enhance clarity and readability.

**Questions For Authors:**

Refer to the weakness

**Relation To Broader Scientific Literature:**

This paper proposes a new peer-review trustworthy information bottleneck method, which is the key contribution of this manuscript.

**Theoretical Claims:**

I have checked the correctness of proofs for theorem 3.3 and its proof in the manuscript and appendix.

---

> ### Author Rebuttal · Authors · 2025-03-31
>
> Thank you for the insightful comments and constructive suggestions. We have carefully revised the whole manuscript and provided detailed responses to each point below.
>
> **1. Generally, in a peer-review process, an EiC is also involved. I am interesting about which part stands for the EiC role in the clustering process.**
>
> ***Response:*** Thanks for your comments. Indeed, an EIC is also involved in an academic peer-review process and we argue that both the EIC and AE can play the same supervisory role. This is consistent with our intention that use the final clustering result to ensure the trustworthiness of the peer-review score in a self-supervision fashion. In the submitted manuscript, AE is mentioned to help readers better understand how to self-supervise and evaluate the trustworthiness of the multimodal peer-review process. In summary, the final clustering result can be regarded as EIC/AE.
>
> **2. The author should detail the ‘k-means-like draw-and-merger algorithm’ in Sec 3.5 using one or two sentences to enhance clarity and readability.**
>
> ***Response:*** Thanks for your comments. The detail of the ‘k-means-like draw-and-merger algorithm’ is that Each sample is sequentially drawn from the old cluster and assigned to an optimal new cluster that minimizes the merger cost to maximize the objective function.
> In the following, we will outline the draw-and-merger algorithm and the optimization process of k-means, demonstrating why the draw-and-merger algorithm works like k-means.
> The optimization process of $k$-means mainly includes the following steps:
>
> (a) Initialize $k$ centroids by randomly selecting $k$ samples.
>
> (b) Assign each data point to the optimal new cluster corresponding to the nearest centroid, which will reduce the $k$-means loss (i.e. Sum of Squared Errors, SSE), and recalculate the centroid of each cluster.
>
> (c) Loop through step (b) until convergence or the maximum number of iterations is reached.
>
> The draw-and-merger algorithm mainly includes the following steps:
>
> (1) Initialize $k$ clusters by randomly assigning samples.
>
> (2) Sequentially reassign each data point to the optimal new cluster corresponding to the minimum merger cost.
>
> (3) Loop through step (2) until convergence or the maximum number of iterations is reached.
>
> Obviously, the steps of both algorithms are similar, and their key step is to reassign each data point to the optimal new cluster. The difference is that the draw-and-merger algorithm reduces the computational complexity by formalizing the merging loss.
>
> Thanks again for the valuable suggestions provided by the reviewer. The modifications will be added to the final version.

---

### Decision · Program_Chairs · 2025-05-01

**Decision:**

Accept (poster)

**Comment:**

This paper focuses on three issues in multi-modal clustering, i.e., lack of trustworthiness in learned weights, isolated view weight learning, extra weight parameters. To solve these issues, motivated by the peer-review mechanism, the authors treat one modality as author and the remaining modalities as reviewers to reach a peer-review score. Then, to boost the trustworthiness, the authors devise a trustworthy score with self-supervision mechanism. Further, they take into account the above scores for accurate and parameter-free modality weight learning.
Experimental results demonstrate the effectiveness of the present algorithm.


After rebuttal, all reviewers give positive scores for this manuscript and acknowledge its contributions. After reading this paper and the posted responses, I agree with the reviewers and recommend an acceptance for this paper.